# The Transport and Uptake of Resveratrol Mediated via Glucose Transporter 1 and Its Antioxidant Effect in Caco-2 Cells

**DOI:** 10.3390/molecules28124569

**Published:** 2023-06-06

**Authors:** Zhen-Dong Zhang, Qi Tao, Li-Xia Bai, Zhe Qin, Xi-Wang Liu, Shi-Hong Li, Ya-Jun Yang, Wen-Bo Ge, Jian-Yong Li

**Affiliations:** 1Key Lab of New Animal Drug Project of Gansu Province, Key Lab of Veterinary Pharmaceutical Development of Ministry of Agriculture and Rural Affairs, Lanzhou Institute of Husbandry and Pharmaceutical Sciences of CAAS, Lanzhou 730050, China; 13027721013@163.com (Z.-D.Z.); taoqi19951224@163.com (Q.T.); bailx552369@163.com (L.-X.B.); qinzhe@caas.cn (Z.Q.); xiwangliu@126.com (X.-W.L.); lzlishihong@163.com (S.-H.L.); yangyue10224@163.com (Y.-J.Y.); gewb1993@163.com (W.-B.G.); 2College of Life Sciences, South China Agricultural University, Guangzhou 510642, China

**Keywords:** resveratrol, Caco-2 cell, uptake, transport, metabolomics

## Abstract

Resveratrol has anti-inflammatory, anti-cancer, and anti-aging pharmacological activities. There is currently a gap in academic research regarding the uptake, transport, and reduction of H_2_O_2_-induced oxidative damage of resveratrol in the Caco-2 cell model. This study investigated the role of resveratrol in the uptake, transport, and alleviation of H_2_O_2_-induced oxidative damage in Caco-2 cells. In the Caco-2 cell transport model, it was observed that the uptake and transport of resveratrol (10, 20, 40, and 80 μM) were time dependent and concentration dependent. Different temperatures (37 °C vs. 4 °C) could significantly affect the uptake and transportation of resveratrol. The apical to basolateral transport of resveratrol was markedly reduced by STF-31, a GLUT1 inhibitor, and siRNA intervention. Furthermore, resveratrol pretreatment (80 μM) improves the viability of Caco-2 cells induced by H_2_O_2_. In a cellular metabolite analysis combined with ultra-high performance liquid chromatography-tandem mass spectrometry, 21 metabolites were identified as differentials. These differential metabolites belong to the urea cycle, arginine and proline metabolism, glycine and serine metabolism, ammonia recycling, aspartate metabolism, glutathione metabolism, and other metabolic pathways. The transport, uptake, and metabolism of resveratrol suggest that oral resveratrol could prevent intestinal diseases caused by oxidative stress.

## 1. Introduction

In humans and animals, the gastrointestinal tract digests food and absorbs nutrients [1]. After oral administration, many Chinese herbal extracts and Chinese herbal monomers will be absorbed by the small intestine after passing through the stomach and large intestine [2,3]. However, intestinal absorption and bioavailability are also affected by a variety of factors (uptake, absorption time, and apparent permeation coefficient, Papp) [4,5]. The apparent permeability coefficient (Papp) reflects the ability of drugs to penetrate monolayer cells and the speed of absorption. There are many cell models used because they are reproducible, cheap, and have short cycle times. Drug transport and absorption studies are increasingly using in vitro models, such as the human colon adenocarcinoma monolayer model and the human umbilical vein endothelial cell model [6,7]. Particularly, the Caco-2 cell model correlates well with in vivo studies and is effective for quick and early drug screening [8]. Intestinal absorption and drug absorption pathways can be determined by calculating the Papp value. The cell monolayer model was used to determine the bidirectional transportation permeability (apical side to basolateral side and basolateral side to apical side, hereinafter referred to as AP–BL and BL–AP) [9,10].

Resveratrol is a natural polyphenol found in red grapes, peanuts, berries, and pomegranates [11,12]. Studies have shown that resveratrol has antioxidant, anti-inflammatory, anti-cardiovascular, anti-apoptotic, anti-diabetic, anti-neurodegenerative, anti-aging, and anti-cancer effects [13,14,15]. The functional properties of resveratrol also make it a popular nutritional supplement [16]. Furthermore, resveratrol has been reported to have cardioprotective properties against ischemia-reperfusion injury in animal models [17]. Resveratrol protects PC12 cells from oxidative damage caused by high glucose via activating the PI3K/Akt signaling pathways [18]. Furthermore, resveratrol inhibited colon cancer cell proliferation via upregulating BMP7 and inhibiting PI3K/Akt signaling [19].

In the present study, the transport and uptake of resveratrol were investigated by successfully constructing a Caco-2 cell monolayer culture model. The antioxidant effect of resveratrol was investigated by the H_2_O_2_-induced Caco-2 cell oxidative damage model. Evidence was presented that resveratrol is absorbed, transported, and acts as an antioxidant in the gut.

## 2. Results

### 2.1. Effects of Different Concentrations of Resveratrol on the Viability of Caco-2 Cells

After incubating the Caco-2 cells with different concentrations of resveratrol (10, 20, 40, and 80 μM) for 24 h, a CCK-8 kit was used to detect the viability of the Caco-2 cells. The results showed that resveratrol had no significant effect on the activity of the Caco-2 cells at 24 h (Figure 1).

### 2.2. Transport Experiment Results

The transport of resveratrol gradually increased with time in the Caco-2 cell model at 37 °C (Figure 2A). The effect of temperature on resveratrol transport was studied at 37 °C and 4 °C. A significant reduction in transport volume was observed at 4 °C compared to 37 °C in the Caco-2 cells (Figure 2B). As the concentration of resveratrol in the Caco-2 cells increased, the number of transporters increased (Figure 3A). On the BL–AP and AP-BL sides, the transmembrane transport rates were concentration dependent at 37 °C (Figure 3B). The BL–AP side of the Caco-2 cells transported resveratrol at 8.57% under 37 °C, and when comparing BL–AP to AP–BL, the percentages were significantly higher (Figure 3C). The above results showed that the transepithelial transport of resveratrol from the AP–BL and BL–AP sides were time dependent, and temperature significantly affects the transport capacity.

### 2.3. Papp of the Transport of Resveratrol

Some studies have shown that the Papp value of drugs or additives can reflect the absorption capacity of substances in the intestine [20]. During the transport of resveratrol at different concentrations on the AP–BL side, the Papp value increased with time and concentration (Figure 4A). In the transport process of the BL–AP side, the Papp values of each concentration group decreased gradually with the increase in time, and the Papp values of each time group decreased gradually with the increase in concentrations from 20~80 μM (Figure 4B). The results indicated the existence of the energy-consuming active transport of resveratrol in the Caco-2 cells during transport.

### 2.4. Uptake Test Results

Resveratrol uptake increased with time in the Caco-2 monolayer cells at 37 °C (Figure 5A). A significant reduction in resveratrol intake is shown in Figure 5B compared to 37 °C. Low temperatures may affect the fluidity of cell membranes, thereby reducing the Caco-2 cells’ intake of resveratrol. In addition, low temperatures can promote cell-to-cell communication by opening up dense connections between cells. Therefore, when the temperature is lowered from 37 °C to 4 °C, the intake of resveratrol by Caco-2 cells will decrease (Figure 5B). When resveratrol was added to Caco-2 monolayer cells under 37 °C and 120 min, the uptake rate decreased with the concentration increase (Figure 5C). The above results showed that resveratrol uptake by Caco-2 cells is time-dependent, and the temperature has an impact on the uptake of resveratrol.

### 2.5. Effect of GLUT1 Inhibition on Resveratrol Transport and Uptake

In order to demonstrate that GLUT1 may be involved in the transport and uptake of resveratrol across the monolayer of Caco-2 cells, we investigated the effects of STF-31 (GLUT1 inhibitor) and siRNA on the transmembrane transport and uptake of resveratrol. Pretreatment of cells with STF-31 and GLUT1 siRNA could significantly reduce the transport of resveratrol (Figure 6A,B). Compared with the control group, the intake of resveratrol was significantly increased in the inhibitor group and the siRNA group (Figure 6C). The results showed that GLUT1 was necessary for the excretion of resveratrol from the Caco-2 cells.

### 2.6. Resveratrol Protects the Cell Viability of H_2_O_2_-Stimulated Caco-2 Cells

In Figure 1, different concentrations of resveratrol had no significant effect on the activity of the Caco-2 cells at 24 h. The activity of the Caco-2 cells pretreated with different concentrations of H_2_O_2_ was significantly decreased. Compared with the control group, the cell viability was reduced to 50.5% after pretreatment of the Caco-2 cells with 500 μM H_2_O_2_ for 3 h (Figure 7A). As shown in Figure 7B, resveratrol was able to effectively increase H_2_O_2_-induced cell viability in a dose-dependent manner.

### 2.7. Metabolomics Analysis of Resveratrol Effect on H_2_O_2_-Induced Caco-2 Cells

Three experimental groups were analyzed with unsupervised PCA in this study. Positive and negative modes explained 86.8% and 72.6% of the total variance, respectively. The PCA plots (Figure 8A,B,G,H) showed clear separation between the three groups in both positive and negative ion modes. We used supervised orthogonal partial least squares discriminant analysis (OPLS–DA) to separate and identify the metabolites further. The OPLS–DA score plots clearly distinguished the H_2_O_2_ group from the other groups without any overlap in either positive or negative modes (Figure 8C,E,I,K). Based on the R2X, R2Y, and Q2 values, the OPLS–DA model was robust and predictive (Figure 8D,F,J,L).

Twenty-one metabolites were identified as potential metabolites, including pipecolinic acid, adenine, uridine, bergenin, glycitin, arginine, citrulline, phosphatidylethanolamine 18, glycine, glutamine, phosphatidylethanolamine 15, threonic acid, methylxanthine, coumaroyl hexoside, malic acid, feruloyl quinic acid, pyroglutamic acid, L-ornithine, L-serine, gluconic acid, and L-glutamic acid (Table 1). As a result of resveratrol treatment, either up- or downregulation of these metabolites occurred.

The metabolic pathway analysis data are shown as a bar chart and a bubble chart in Figure 9. According to the KEGG enrichment analysis of differentially expressed metabolites, 25 differentially expressed metabolite pathways were enriched: urea cycle, arginine and proline metabolism, glycine and serine metabolism, ammonia recycling, aspartate metabolism, glutathione metabolism, etc. (*p* < 0.05). The influence of the path is mainly concentrated in arginine biosynthesis, glutathione metabolism, aminoacyl-tRNA biosynthesis, glyoxylate and dicarboxylate metabolism, arginine and proline metabolism, glycine, serine and threonine metabolism, and D-glutamine and D-glutamate metabolism. The above results show that the oxidative injury induced by H_2_O_2_ caused metabolic disorder, and resveratrol could effectively regulate this imbalance (Figure 10).

## 3. Discussion

Polyphenols are organic compounds found in plant foods [21,22]. Polyphenolic compounds such as resveratrol have many functions [23,24]. Following metabolism, resveratrol is excreted through the urine and feces when consumed orally [25,26]. In the body, resveratrol has relatively low bioavailability. There is approximately 1% bioavailability of resveratrol metabolites in the small intestine and liver. In mammals, resveratrol is metabolized primarily as sulfated and glucuronidated products. A Caco-2 cell model was used in this study to observe the transport and uptake of resveratrol. GLUT1 was found to mediate resveratrol transport in the study. However, some studies have shown that GLUT1 is not necessary for resveratrol transport, but an ATP-binding cassette transporter is necessary [27]. This study found that PappBA polarized resveratrol absorption, and that transporter carriers transported resveratrol. Resveratrol transport was significantly inhibited by STF-31 and GLUT1 siRNA, and it was shown that GLUT1 was most likely involved in resveratrol transport.

As a result of this experiment, resveratrol’s Papp in the Caco-2 cell monolayer ranged from 1.0 × 10^−6^ to 4.0 × 10^−6^ cm·s^−1^. The results could predict the mechanism by which resveratrol is absorbed in the intestinal tract in reality. In accordance with current international judging standards on substance absorption, a transport rate of 89.62% was observed for resveratrol in the Caco-2 cells, and resveratrol was readily absorbed. There is evidence that macromolecules are transported in Caco-2 cells via transcellular or paracellular pathways [28]. Paracellular pathways open tight junctions so molecules can pass between cells. This study found that TEER did not decrease significantly, suggesting that resveratrol rarely crosses cellular membranes via paracellular routes.

Studies have shown that trans-resveratrol crosses the cells’ apical membrane through passive transport, while trans-piceid crosses through active transport [29]. In Caco-2 cells, there is evidence that the multidrug-related protein 2 (MRP2), an efflux pump, plays a role in stilbene efflux. An MRP inhibitor, MK-571, appears to inhibit trans-piceid efflux and trans-resveratrol efflux from the apical membrane [30]. It is possible that resveratrol’s oral bioavailability will be limited as enterocytes efficiently metabolize it, mostly by sulfate conjugation and to a lesser extent by glucuronidation.

Temperature changes can significantly affect the fluidity of cell membranes. When the external temperature decreases significantly, the fluidity of the cell membrane also decreases significantly. At low temperatures, enzyme activity in the free cells slows and the fluidity of the cell membrane is decreased, and this interferes with the transport mechanisms. When the Caco-2 cells were at 37 °C, the transport and uptake of resveratrol were normal. However, when the Caco-2 cells were in a low-temperature (4 °C) environment, the transport and uptake of resveratrol significantly decreased. The decrease in temperature not only affects the fluidity of cell membranes, but it also affects the vitality of cells. When a cell is at a lower temperature, its vitality is also affected. When cell viability decreases, the transport and uptake of drugs also decrease accordingly. In this study, whether the fluidity of cell membranes and cell viability affect uptake and transport remains a scientific issue that we will continue to explore in the future.

In cellular metabolomics, levels of endogenous small molecule metabolites are studied in response to various stimuli in order to detect changes in metabolic pathways. A cellular metabolomic analysis of Caco-2 cells pretreated with resveratrol revealed 21 differential metabolites. These differential metabolites were involved in amino acid metabolism, carbohydrate metabolism, energy metabolism, and the metabolism of cofactors. The arginine, citrulline, serine, and glutamic acid concentrations in the H_2_O_2_ group were lower, while the amino acid concentrations were significantly higher in the resveratrol group. In the urea cycle, citrulline plays an important role [31,32]. As a result of oxidative stress, arginine, synthesized from citrulline, was found to play an important role in the regulation of immune responses [33]. An enhanced arginine–citrulline–nitric oxide pathway reduces the oxidative damage caused by H_2_O_2_. In the H_2_O_2_ group, the adenine levels were significantly higher. In oxidative stress studies, adenine levels are often increased [34,35]. In this case, however, it is also possible that they represent ATP loss as part of tissue turnover [36]. It has been suggested that the increase in ornithine and serine over the course of life can contribute to an increased risk of apoptosis, mitochondrial dysfunction, inflammation, lipid metabolism, autophagy, as well as antioxidant stress [37]. Compared with the control group, the level of bergenin was significantly decreased in the H_2_O_2_ group; however, resveratrol could significantly increase its level. Antioxidant and anti-inflammatory properties have been demonstrated for bergenin in studies [38,39]. Bergenin hinders the production of extracellular matrix in glomerular mesangial cells and mitigates diabetic nephropathy in mice by suppressing oxidative stress through the mTOR/β-TrcP/Nrf_2_ pathway [40]. Through immunomodulatory and antioxidant effects, bergenin protects Balb/c mice against cyclophosphamide-induced immunosuppression [41]. In another study, lipopolysaccharide-induced mastitis was observed in mice; inflammatory effects are exerted by bergenin via regulating MAPK and NF-κB signaling pathways [42]. In the H_2_O_2_ group, pyroglutamic acid and glutamic acid levels were significantly higher. In glutathione metabolism, glutamic acid is an intermediate. An elevated glutamic acid level affects glutathione metabolism [43,44]. According to the data, resveratrol reduced glutamic acid, while H_2_O_2_ increased it, indicating the protective effect of resveratrol in normalizing glutathione metabolism. In the cell membrane, phospholipids called phosphatidylethanolamines play an important role [45,46]. As compared with the control group, the levels of phosphatidylethanolamine 18 and 15 were significantly increased in the H_2_O_2_ group. The phosphatidylethanolamine levels were significantly reduced by resveratrol. The majority of these differential metabolites are involved in phosphatidylethanolamine and phosphatidylcholine biosynthesis. The oxidative stress indicator threonic acid is a product of the degradation of ascorbic acid. In the H_2_O_2_ group, the threonine levels were significantly lower than in the control group. However, after the resveratrol treatment, the threonine level was significantly recovered. In the metabolism of methylxanthine, 7-Methylxanthine is formed as a purine component. 7-methylxanthine may be closely related to resveratrol’s antioxidant and anti-inflammatory properties. A multitude of cellular processes are affected by malic acid, including the glyoxylate and citric acid cycles [47,48]. Resveratrol was able to significantly increase the level of malic acid compared with the H_2_O_2_ group. The results showed that resveratrol can accelerate the citric acid cycle, which speeds up the production of ATP. The glutathione metabolic pathway includes a differential metabolite, pyroglutamate, which is altered by H_2_O_2_ and reversed by resveratrol. In maintaining redox homeostasis, glutathione protects cells against oxidative damage and toxicity from exogenous electrophiles. Glutathione metabolism is impaired by elevated levels of pyroglutamate, an intermediate in glutathione synthesis. Resveratrol decreased the level of this metabolite, which was increased by H_2_O_2_. The results suggest that resveratrol appears to normalize glutathione metabolism in a protective manner. The results of the metabolomics analysis showed that resveratrol might improve H_2_O_2_ induced oxidative damage by increasing the antioxidant level.

Metabolomics analyses using UPLC/MS are limited in their ability to identify many differentially altered metabolites. In the present study, based on the metabolomics analysis conducted with UPLC/MS, the results of the present study explained the biological changes between the groups only partially. There is certainly a need for further research on the interactions between resveratrol and these metabolic pathways.

## 4. Materials and Methods

### 4.1. Chemicals

Transwell TM cell culture dishes were obtained from Corning Costar Corp (Cambridge, MA, USA). The CCK-8 kit was purchased from Beyotime (Shanghai, China). The MEM glucose medium, Hank’s buffer, and cell culture flask were obtained from Gibco (Grand Island, NY, USA). The Lucifer yellow was from Solarbio (Beijing, China). The MS-gradeacetonitrile was purchased from Thermo Fisher Scientific (Waltham, MA, USA). The formic acid (98.0%, for LC-MS) was purchased from Tokyo Chemical Industry (Shanghai, China). The glucose transporter 1 siRNA sequence: 5′-AUUCCAUUCACAGUGC UGAUCGCUC-3′; for the scrambled control siRNA: 5′-UACUCAG AACGAGUCUCGUUT-3′. The siRNA specifically targeting glucose transporter 1 and the control siRNA were purchased from Jikai Gene Chemical (Shanghai, China). The STF-31 was purchased from MCE (Shanghai, China). The resveratrol (purity greater than 99%) was purchased from Merck (Darmstadt, Germany).

### 4.2. Cell Culture

The cell cultures of Caco-2 were grown as before [49]. The Caco-2 cell line, derived from human epithelial colorectal adenocarcinoma, was cultured in Caco-2 cell culture medium. This medium consisted of DMEM high glucose medium supplemented with 10% FBS, 2% L-glutamine, 1% NEAA, and 1% penicillin-streptomycin mix. Before 500 μM H_2_O_2_ was used to intervene in the cells, the Caco-2 cells were pretreated with resveratrol for 24 h.

### 4.3. Cell Viability

Cell viability was determined via CCK-8 assay [50]. The Caco-2 cells were inoculated into 96-well plates and intervened with resveratrol and H_2_O_2_. After 24 h, 10 µL of CCK-8 solution was added to each well. After 2 h, the optical density value of each hole at 450 nm was measured using a microplate reader (Bio-Rad 550, Hercules, CA, USA).

### 4.4. Establishment and Evaluation of the Caco-2 Cell Uptake Model

As previously published, the Caco-2 cell model was established [49]. In brief, the Caco-2 cells were seeded onto polycarbonate 6-well Transwell inserts at a density of 2 × 10^6^ cells/well; then, the confluent monolayers (21 days) were used for permeability studies. The transepithelial electrical resistance (TEER) of the monolayer was measured using an epithelial voltohmmeter (World Precision Instruments, Sarasota, FL, USA) to determine the formation of the monolayer and its integrity during the experiment. The monolayers were washed with Hank’s balanced salt solution (HBSS) prior to the experiment, after which 0.5 and 1.2 mL of HBSS were placed into the upper and lower compartments, respectively. A total volume of 100 μL of solution was taken from the lower compartment at regular intervals over 120 min and replaced with the same volume of fresh buffer.

### 4.5. Analytical Methods

#### 4.5.1. HBSS Buffer Sample Extraction

Approximately 250 μL HBSS buffer sample containing resveratrol was collected, and 250 μL methanol was added into it and mixed evenly. The mixture was centrifuged in a 14,000× *g* centrifuge for 10 min. After collecting and evaporating the supernatant, it was reconstituted in 200 μL methanol and analyzed using ultra-high liquid chromatography (UHPLC).

#### 4.5.2. Liquid Chromatography Analysis

The resveratrol was analyzed on an Agilent 1290 UHPLC system (Agilent Technologies, Waldbronn, Germany). In the mobile phase, 0.2% phosphoric acid and methanol (40:60, *v*/*v*) was combined at a flow rate of 1 mL/min in an acid solution. A constant wavelength of 306 nm was used to record resveratrol elution.

### 4.6. Transport Assay of Caco-2 Cells

The AP and BL sides were tested separately as part of the resveratrol transport experiment. Different concentrations of resveratrol solutions (5, 10, 20, 40, and 80 μM) were added to the cells. Approximately 250 μL of sample solution was collected on the BL side at different times. Approximately 250 μL of HBSS was added immediately. We investigated the effect on the transport rate of resveratrol at 37 °C and 4 °C, respectively. 

### 4.7. Caco-2 Cells Uptake Experiment

Different concentrations of resveratrol solution (5, 10, 20, 40, and 80 μM) were added to the cells. The cells were collected at different time periods. Similar to the above steps, the effect of temperature (4 °C) on resveratrol uptake was studied.


(1)
The uptake rate (%) = (Resveratrol content in cells/Total amount of resveratrol) × 100%.


### 4.8. Metabonomic Analysis

#### 4.8.1. Cellular Metabolite Extraction

The Caco-2 cells were treated with resveratrol and H_2_O_2_ and collected into centrifuge tubes. After, 80% methanol was added into the centrifuge tube; the sample was mixed and frozen at −80 °C for 5 min. The mixture was lysed with two freeze–thaw cycles (frozen in liquid nitrogen and thawing at 37 °C). Quality control (QC) samples were prepared from 20 μL of supernatant from each sample.

#### 4.8.2. UHPLC–QE Orbitrap/MS/MS Conditions

An HPLC system coupled to an Orbitrap MS was used for the LC–MS/MS analysis. In mobile phase A, 0.1% formic acid was added to water, while in mobile phase B, acetonitrile was added. In this experiment, the elution gradient was set as follows: 0 min, 2% B; 1 min, 2% B; 18 min, 100% B; 22 min, 100% B; 22.1 min, 2% B; and 25 min, 2% B. The flow rate was 0.3 mL/min. The injection volume was 2 μL. During the LC/MS experiments, a QE mass spectrometer was used for acquiring MS/MS spectra on an information-dependent basis (IDA). The ESI source conditions were as follows: the aux gas flow rate as 16 Arb, the collision energy as 25 eV in the NCE model, and the spray voltage as −3.0 kV or 3.6 kV.

#### 4.8.3. Qualitative Analysis of Metabolites

ABF Converter software was used to convert the raw data into “Analysis Base File” (ABF) files. In the data processing, MSDIAL 2.2.62 software was used to detect peaks, deconvolve them, and align them. SIMCA (version 14.1, Umetrics AB, Umeå, Sweden) was used to import the data obtained. A PCA and an OPLS–DA analysis was performed on the SIMCA data. Mass values of differential metabolites were retrieved from the Human Metabolome Database (HMDB). MetabolAnalyst 5.0 was used to analyze differential metabolites in clusters and pathway analyses.

### 4.9. Statistical Analysis

The data were analyzed using GraphPad Prism 8 (Graphpad Software, LaJolla, CA, USA). Means and standard deviations were presented for all data. In addition to one-way ANOVA, Duncan’s multiple comparisons was used to analyze differences between the treatment groups. Statistical significance was considered at *p* < 0.05.

## 5. Conclusions

In the Caco-2 cell model, resveratrol is transported and absorbed mainly by GLUT1. The results of cell metabolomics show that resveratrol can play an important role in the prevention of intestinal diseases caused by oxidative stress as an additive.

## Figures and Tables

**Figure 1 molecules-28-04569-f001:**
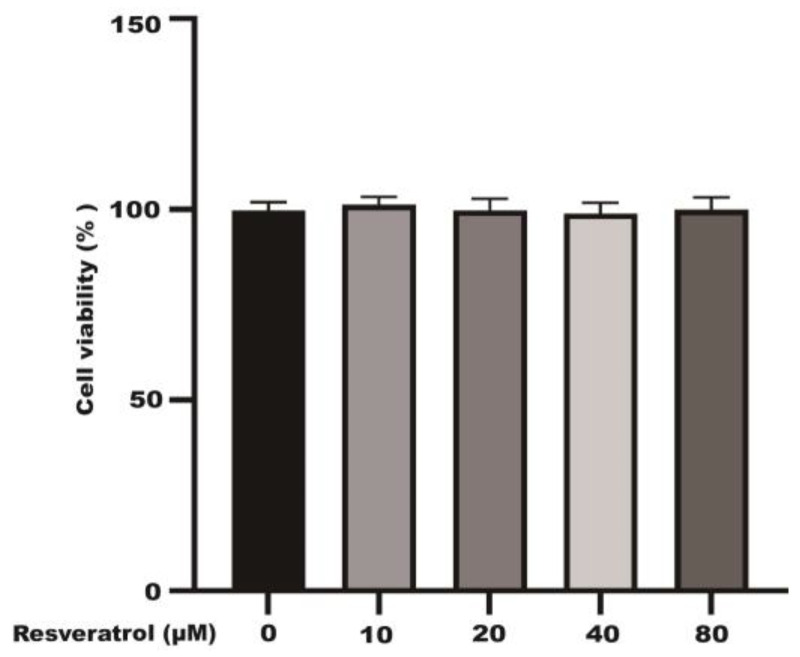
The effects of different concentrations of resveratrol (10~80 μM) on Caco-2 cell viability.

**Figure 2 molecules-28-04569-f002:**
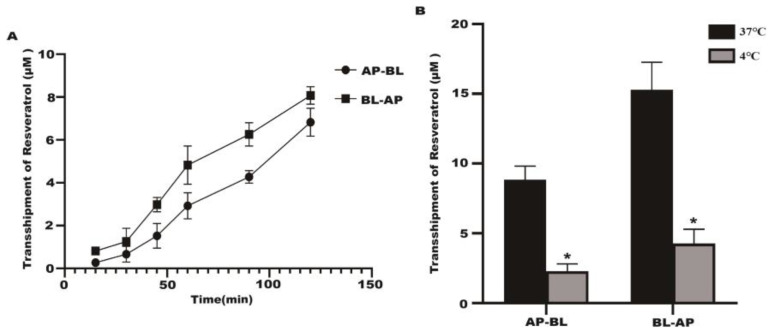
The transport of resveratrol at different times and temperature. (**A**) The transport of AP–BL and BL–AP of resveratrol (80 μM) at different times. (**B**) The transport of AP–BL and BL–AP of resveratrol at different temperatures. * *p* < 0.05 compared with 37 °C.

**Figure 3 molecules-28-04569-f003:**
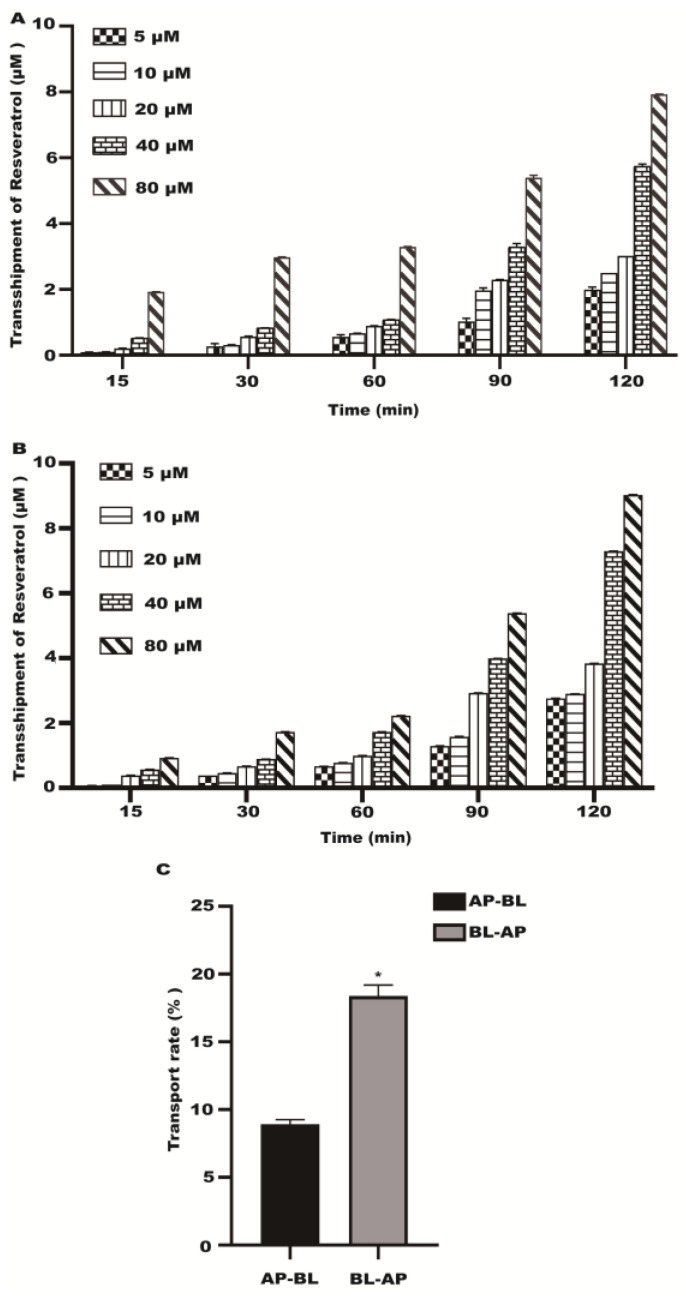
The transport of resveratrol with different concentrations (5~80 μM). (**A**) The transport of resveratrol in the AP–BL. (**B**) The transport of resveratrol in the BL–AP. (**C**) The transport rate of resveratrol. * *p* < 0.05 compared with AP-BL side.

**Figure 4 molecules-28-04569-f004:**
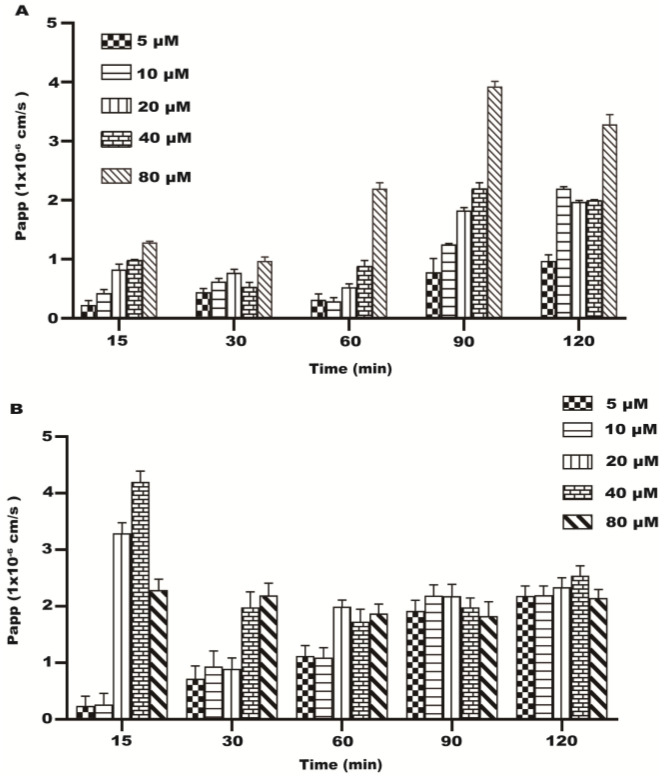
The Papp of each concentration (5~80 μM) of resveratrol during 120 min. (**A**) The Papp of AP–BL. (**B**) The Papp of BL–AP.

**Figure 5 molecules-28-04569-f005:**
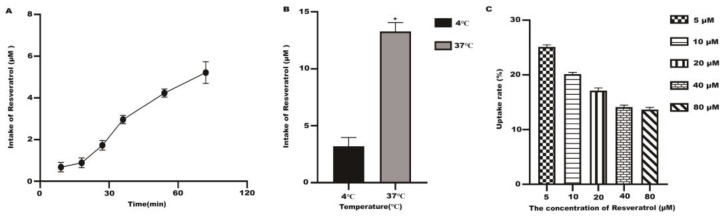
The uptake of resveratrol with different concentrations (5~80 μM). (**A**) The uptake of resveratrol (80 μM) at different times. (**B**) The uptake of resveratrol at different temperatures. (**C**) The uptake rate of resveratrol of Caco-2 cells. * *p* < 0.05 compared with 4 °C.

**Figure 6 molecules-28-04569-f006:**
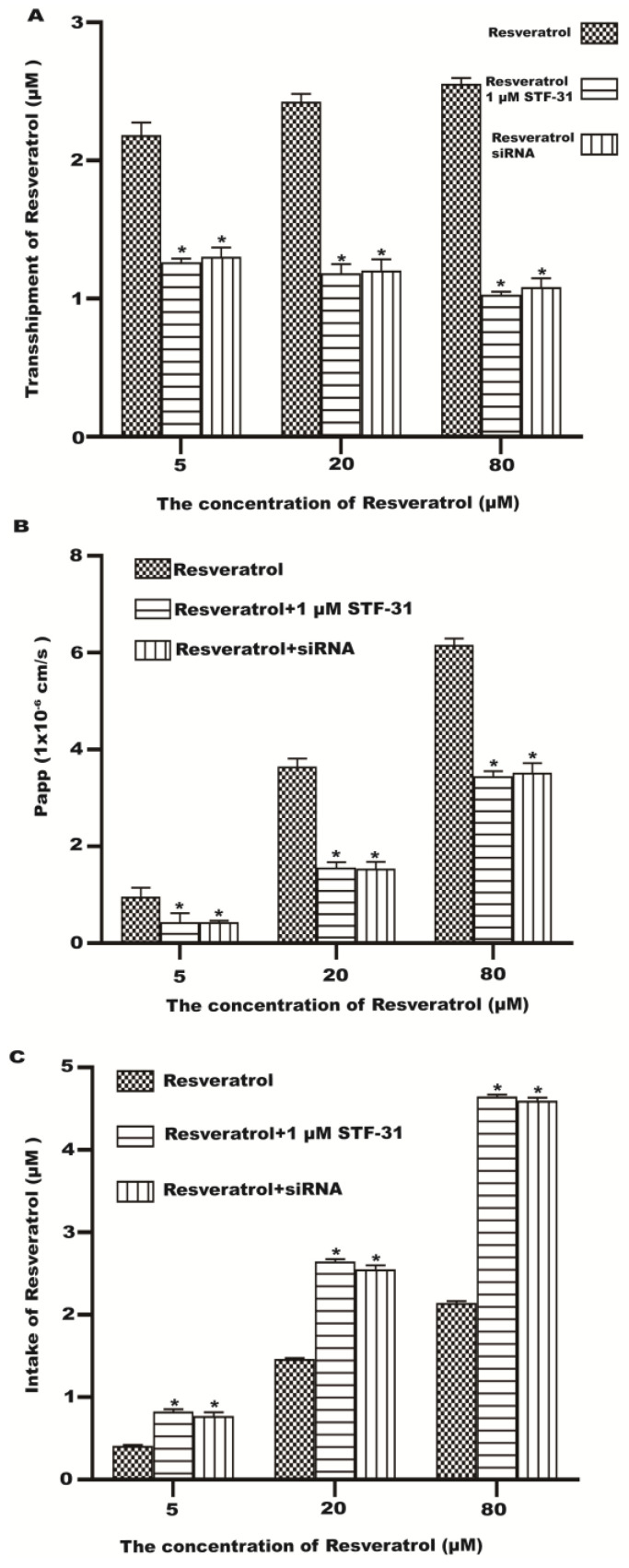
The effect of STF-31 and siRNA on resveratrol transport and uptake in Caco-2 cells. (**A**) The transport of different concentrations of resveratrol. (**B**) The Papp of each concentration of resveratrol. (**C**) The intake of different concentrations of resveratrol. * *p* < 0.05 compared with the resveratrol group.

**Figure 7 molecules-28-04569-f007:**
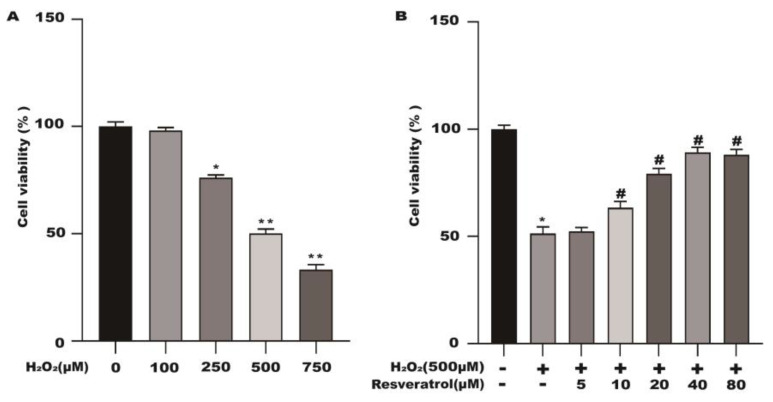
Resveratrol protects the cell viability of H_2_O_2_-stimulated Caco-2. (**A**) Effects of different concentrations of H_2_O_2_ on cell viability. (**B**) Effects of different concentrations of resveratrol pretreatment on H_2_O_2_-induced cell viability. * *p* < 0.05 compared with the control group; ** *p* < 0.01 compared with the control group; *# p* < 0.05 compared with the H_2_O_2_ group.

**Figure 8 molecules-28-04569-f008:**
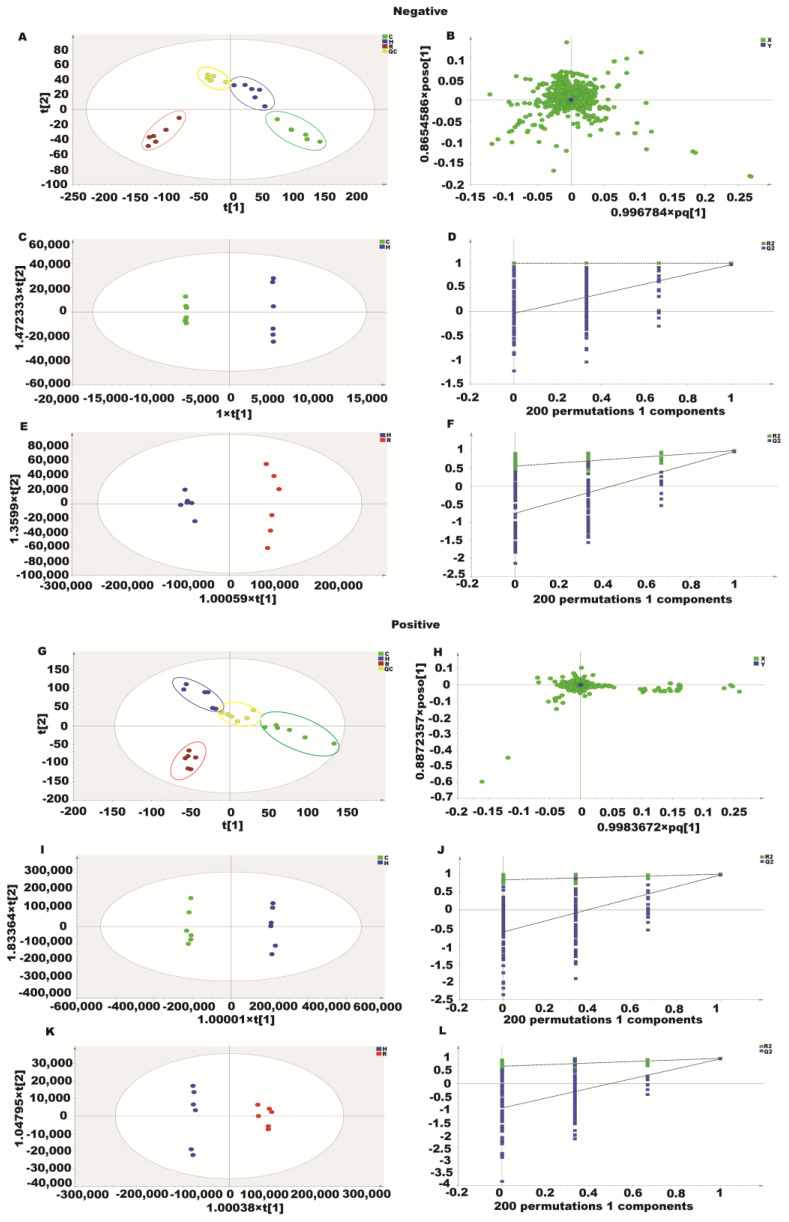
Metabolomics analysis of the effect of resveratrol on H_2_O_2_-induced oxidative injury in Caco-2 cells. (**A**,**G**) PCA score plots based on the Caco-2 cells of the control, H_2_O_2_ and resveratrol groups in positive and negative modes. (**B**,**H**) The loading plot of resveratrol and H_2_O_2_ groups in positive and negative modes. (**C**,**I**) OPLS–DA score plots of the resveratrol and H_2_O_2_ groups in positive and negative modes. (**D**,**J**) Permutation test of the OPLS–DA model. (**E**,**K**) OPLS–DA score plots of the control and H_2_O_2_ groups in positive and negative modes. (**F**,**L**) Permutation test of the OPLS–DA model.

**Figure 9 molecules-28-04569-f009:**
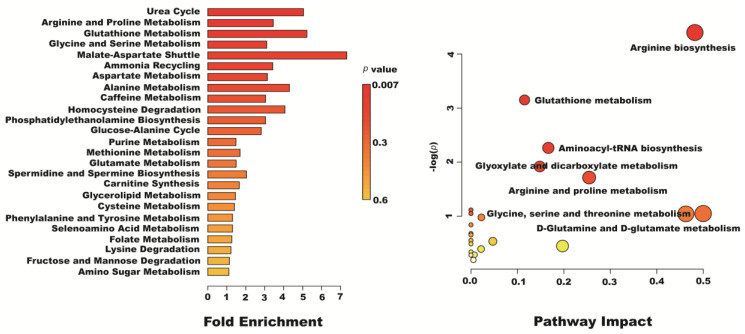
The results of fold enrichment and pathway analysis of potential metabolites in Caco-2 cells.

**Figure 10 molecules-28-04569-f010:**
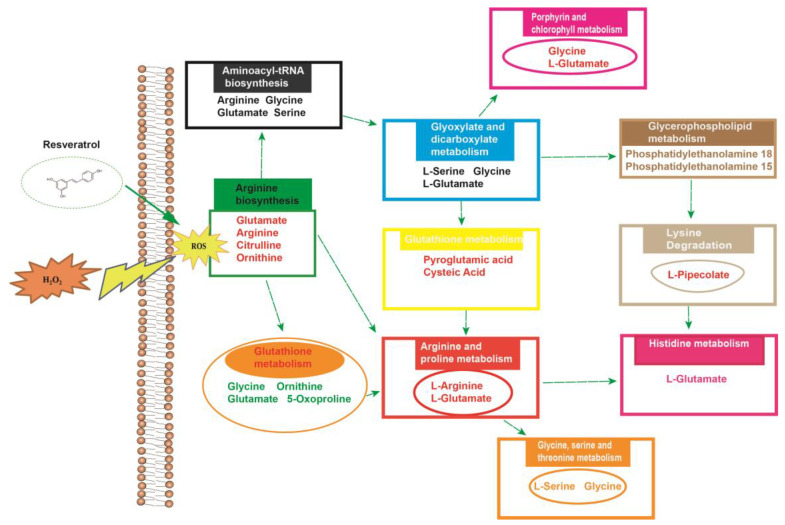
Overview of altered concentrations of metabolites and related metabolic pathways.

**Table 1 molecules-28-04569-t001:** Statistics of differential metabolites in Caco-2 cells.

No.	RT	VIP	Formula	Metabolites	SM	*m*/*z*	Fold Change	
							C/H	H/R
1	1.072532	3.38	C_6_H_11_NO_2_	Pipecolinic acid	ESI+	129.157	0.42	2.91
2	7.066492	1.31	C_5_H_5_N_5_	Adenine	ESI+	680.344	1.38	2.42
3	1.835533	2.55	C_9_H_12_N_2_O_6_	Uridine	ESI+	245.0759	1.21	2.95
4	4.238871	3.65	C_14_H_16_O_9_	Bergenin	ESI+	328.27	0.68	2.56
5	0.913932	4.42	C_22_H_22_O_10_	Glycitin	ESI+	485.015	0.63	4.27
6	0.947492	1.86	C_6_H_14_N_4_O_2_	Arginine	ESI+	174.20	0.22	3.83
7	0.918416	5.23	C_6_H_13_N_4_O_3_	Citrulline	ESI+	193.131	0.41	2.36
8	8.736221	2.23	C_40_H_80_NO_8_P	Phosphatidylethanolamine 18	ESI+	734.039	1.58	0.84
9	1.436667	1.38	C_2_H_5_NO_2_	Glycine	ESI+	76.03984	1.85	4.57
10	1.576442	1.51	C_10_H_20_N_4_O_6_	Glutamine	ESI+	185.029	0.84	2.48
11	3.923614	2.31	C_41_H_78_NO_8_P	Phosphatidylethanolamine 15	ESI+	744	1.38	0.14
12	4.836229	1.33	C_8_H_14_CaO_10_	Threonic acid	ESI+	310.27	0.59	2.84
13	1.880109	2.41	C_6_H_6_N_4_O_2_	7-Methylxanthine	ESI+	167.0549	2.78	0.89
14	5.302783	3.02	C_9_H_3_C_l6_NO_2_	Coumaroyl Hexoside	ESI−	369.83	0.52	1.83
15	1.376172	3.06	C_4_H_6_O_5_	Malic acid	ESI−	135.0273	0.72	2.88
16	4.237209	3.17	C_17_H_20_O_9_	Feruloyl quinic acid	ESI−	368.3353	0.27	2.98
17	0.994167	2.44	C_5_H_7_NO_3_	Pyroglutamic acid	ESI−	130.0497	2.67	0.66
18	0.582613	2.82	C_5_H_13_N_2_O_2_	L-ornithine	ESI−	132.16	0.78	2.86
19	0.996538	4.25	C_3_H_7_NO_3_	L-serine	ESI−	106.05	0.97	3.75
20	1.862335	2.99	C_6_H_11_O_7_	Gluconic acid	ESI−	195.148	0.42	1.56
21	1.641869	1.72	C_5_H_9_NO_4_	L-Glutamic acid	ESI−	148.0603	0.86	2.52

RT: retention time; VIP: variable importance in the projection; SM: scan mode; +: metabolites identified in positive mode; −: metabolites identified in negative mode. Metabolites identified in both positive and negative modes; C/H: control group compared with the H_2_O_2_ group; R/H: resveratrol group compared with the H_2_O_2_ group.

## Data Availability

The data used to support the findings of this study are included within the article.

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
