# Peer review of "The Transport and Uptake of Resveratrol Mediated via Glucose Transporter 1 and Its Antioxidant Effect in Caco-2 Cells"

_molecules, 2023, doi:10.3390/molecules28124569_

Round 1

Reviewer 1 Report

The study entitled “The transport and uptake of resveratrol mediated via glucose 2 transporter 1 and its antioxidant effect in Caco-2 cells”. Please find the comments below:

1.    Line 14-15 “However, there are few studies on the absorption, transportation and antioxidant effects of resveratrol in vitro”

The authors should emphasize other research gaps since there are various studies being done on the absorption, transportation and antioxidant effects of resveratrol in vitro.

2.    Specify the dosage for Resveratrol pretreatment in the abstract (Line 20).

3.    Line 35: The abbreviation “Papp” should be explained the first time it is used.

4.    The methodology section should be presented in more detail to allow reproducibility.

5.    What is the significance of carrying out the analysis at 2 different temperatures (4⁰C and 37⁰C)?

6.    Line 79: “250 μL sample was collected”. Which sample? This should be clarified.

7.    How long did the Caco2 cells pre-treated with resveratrol prior to the addition of 500 μM H2O2? The methods should be clearly explained.

8.    How and why does the dose 0-80 uM of resveratrol chosen?

9.    Line 262: “In grape glomerular mesangial cells…” needs more clarification.

10. What is the statistical significance of Figures 3A and 3B? Similarly for Figures 4A, 4B and 5C.

11. References need to be updated as many of them are more than 5 years.

12. The effect of temperatures on resveratrol uptake is not discussed at all in the discussion section.

13. The manuscript has around 35% of similarity. Suggest to revise and reducing the percentage.

Languages and formatting should be checked and improved throughout the manuscript

Eg: Line 37, Line 64: MS-gradeacetonitrile

Line 66-68:The sentence “Glucose transporter 1 siRNA sequence……” is hanging.

Author Response

Dear Reviewers,

Thank you for the reviewer’ comments concerning our manuscript entitled "The transport and uptake of resveratrol mediated via glucose transporter 1 and its antioxidant effect in Caco-2 cells" (Manuscript ID molecules-2425652). Those comments were all valuable and very helpful for revising and improving our paper, as well as for providing important guiding significance to our study. We have studied the comments carefully and have made corrections, which we hope will meet with your approval. The revised text is marked in yellow in the revised version of the manuscript. The main corrections made to the paper and the responses to the reviewer’ comments with point by point are as follows:

Reviewer #1:

Comments and Suggestions for Authors

The study entitled “The transport and uptake of resveratrol mediated via glucose 2 transporter 1 and its antioxidant effect in Caco-2 cells”. Please find the comments below:

  1. Line 14-15 “However, there are few studies on the absorption, transportation and antioxidant effects of resveratrol in vitro”. The authors should emphasize other research gaps since there are various studies being done on the absorption, transportation and antioxidant effects of resveratrol in vitro.

Thank you very much for the suggestions provided by the reviewers. We have made modifications to this sentence.

“There is currently a gap in academic research regarding the uptake, transport, and reduction of H2O2-induced oxidative damage of resveratrol in the Caco-2 cell model.”

  1. Specify the dosage for Resveratrol pretreatment in the abstract (Line 20).

We have added the dosage of resveratrol (80 μM) to the manuscript abstract (Line 20).

  1. Line 35: The abbreviation “Papp” should be explained the first time it is used.

We have annotated Papp (apparent permeation coefficient) when it first appeared.

“The apparent permeability coefficient (Papp) reflects the ability of drugs to penetrate monolayer cells and the speed of absorption.”

  1. The methodology section should be presented in more detail to allow reproducibility.

We have provided a more detailed supplement to the methodology section of the manuscript.

  1. What is the significance of carrying out the analysis at 2 different temperatures (4℃ and 37℃)?

Studies have shown that different temperatures have a significant impact on drug transport and uptake. Therefore, we referred to some articles and designed two temperatures (4℃ and 37℃).

References

[1] Lu J, Liu L, Zhu X, Wu L, Chen Z, Xu Z, Li W. Evaluation of the Absorption Behavior of Main Component Compounds of Salt-Fried Herb Ingredients in Qing'e Pills by Using Caco-2 Cell Model. Molecules. 2018 Dec 14;23(12):3321. doi: 10.3390/molecules23123321.

[2] Srisongkram T, Weerapreeyakul N, Kärkkäinen J, Rautio J. Role of L-Type Amino Acid Transporter 1 (LAT1) for the Selective Cytotoxicity of Sesamol in Human Melanoma Cells. Molecules. 2019 Oct 27;24(21):3869. doi: 10.3390/molecules24213869.

[3] Zhang ZD, Tao Q, Qin Z, Liu XW, Li SH, Bai LX, Yang YJ, Li JY. Uptake and Transport of Naringenin and Its Antioxidant Effects in Human Intestinal Epithelial Caco-2 Cells. Front Nutr. 2022 May 24;9:894117. doi: 10.3389/fnut.2022.894117.

  1. Line 79: “250 μL sample was collected”. Which sample? This should be clarified.

250 μL HBSS buffer sample containing resveratrol were collected and 250 μL methanol was added into it and mixed evenly.

  1. How long did the Caco2 cells pre-treated with resveratrol prior to the addition of 500 μM H2O2? The methods should be clearly explained.

Before 500 μM H2O2 was used to intervene in cells, Caco-2 cells were pretreated with resveratrol for 24 h.

We have added this sentence to the methodology section of the manuscript.

  1. How and why does the dose 0-80 μM of resveratrol chosen?

We conducted some preliminary experiments on the dosage of resveratrol. We found that the dosage of resveratrol (0-80 μM) effectively alleviated H2O2 induced oxidative damage to cells without affecting cell viability.

  1. Line 262: “In grape glomerular mesangial cells…” needs more clarification.

We have further clarified and elaborated on this sentence.

“Bergenin hinders the production of extracellular matrix in glomerular mesangial cells and mitigates diabetic nephropathy in mice by suppressing oxidative stress through the mTOR/β-TrcP/Nrf2 pathway.”

  1. What is the statistical significance of Figures 3A and 3B? Similarly for Figures 4A, 4B and 5C.

The statistical significance of Figures 3A and 3B was to demonstrate that different concentrations of resveratrol significantly increase their transport over time.

Figure 4A showed that the Papp value increased with time and concentration during the transport of resveratrol at different concentrations on the AP-BL side. Figure 4B showed that Papp values showed an opposite trend during the transport of resveratrol at different concentrations on the BL-AP side.

Figure 5C showed that the uptake rate of resveratrol by cells decreases with increasing concentration.

  1. References need to be updated as many of them are more than 5 years.

We have updated the references.

References

Jacobson A, Yang D, Vella M, Chiu IM: The intestinal neuro-immune axis: crosstalk between neurons, immune cells, and microbes. Mucosal Immunol 2021, 14:555-565.

Gao Y, Liang Z, Lv N, Shan J, Zhou H, Zhang J, Shi L: Exploring the total flavones of Abelmoschus manihot against IAV-induced lung inflammation by network pharmacology. BMC Complement Med Ther 2022, 22:36.

Kriseldi R, Johnson JA, Walk CL, Bedford MR, Dozier WA, 3rd: Influence of exogenous phytase supplementation on phytate degradation, plasma inositol, alkaline phosphatase, and glucose concentrations of broilers at 28 days of age. Poult Sci 2021, 100:224-234.

Gianchecchi E, Fierabracci A: Insights on the Effects of Resveratrol and Some of Its Derivatives in Cancer and Autoimmunity: A Molecule with a Dual Activity. Antioxidants (Basel) 2020, 9.

Carr AC, Rowe S: Factors Affecting Vitamin C Status and Prevalence of Deficiency: A Global Health Perspective. Nutrients 2020, 12.

Efenberger-Szmechtyk M, Nowak A, Nowak A: Cytotoxic and DNA-Damaging Effects of Aronia melanocarpa, Cornus mas, and Chaenomeles superba Leaf Extracts on the Human Colon Adenocarcinoma Cell Line Caco-2. Antioxidants (Basel) 2020, 9.

Aleksiejczuk M, Gromotowicz-Poplawska A, Marcinczyk N, Stelmaszewska J, Dzieciol J, Chabielska E: Aldosterone Increases Vascular Permeability in Rat Skin. Cells 2022, 11.

Cui Y, Claus S, Schnell D, Runge F, MacLean C: In-Depth Characterization of EpiIntestinal Microtissue as a Model for Intestinal Drug Absorption and Metabolism in Human. Pharmaceutics 2020, 12.

Morresi C, Vasarri M, Bellachioma L, Ferretti G, Degl Innocenti D, Bacchetti T: Glucose Uptake and Oxidative Stress in Caco-2 Cells: Health Benefits from Posidonia oceanica (L.) Delile. Mar Drugs 2022, 20.

Arena A, Romeo MA, Benedetti R, Masuelli L, Bei R, Gilardini Montani MS, Cirone M: New Insights into Curcumin- and Resveratrol-Mediated Anti-Cancer Effects. Pharmaceuticals (Basel) 2021, 14.

Summerlin N, Soo E, Thakur S, Qu Z, Jambhrunkar S, Popat A: Resveratrol nanoformulations: challenges and opportunities. Int J Pharm 2015, 479:282-290.

Sales JM, Resurreccion AV: Resveratrol in peanuts. Crit Rev Food Sci Nutr 2014, 54:734-770.

de Oliveira MR, Nabavi SF, Manayi A, Daglia M, Hajheydari Z, Nabavi SM: Resveratrol and the mitochondria: From triggering the intrinsic apoptotic pathway to inducing mitochondrial biogenesis, a mechanistic view. Biochim Biophys Acta 2016, 1860:727-745.

Wang X, Xie Y, Zhang T, Bo S, Bai X, Liu H, Li T, Liu S, Zhou Y, Cong X, et al: Resveratrol reverses chronic restraint stress-induced depression-like behaviour: Involvement of BDNF level, ERK phosphorylation and expression of Bcl-2 and Bax in rats. Brain Res Bull 2016, 125:134-143.

Kalantari H, Das DK: Physiological effects of resveratrol. Biofactors 2010, 36:401-406.

Frazzi R, Tigano M: The multiple mechanisms of cell death triggered by resveratrol in lymphoma and leukemia. Int J Mol Sci 2014, 15:4977-4993.

Ling L, Gu S, Cheng Y: Resveratrol activates endogenous cardiac stem cells and improves myocardial regeneration following acute myocardial infarction. Mol Med Rep 2017, 15:1188-1194.

Liu MH, Yuan C, He J, Tan TP, Wu SJ, Fu HY, Liu J, Yu S, Chen YD, Le QF, et al: Resveratrol Protects PC12 Cells from High Glucose-Induced Neurotoxicity Via PI3K/Akt/FoxO3a Pathway. Cellular and Molecular Neurobiology 2015, 35:513-522.

Zeng YH, Zhou LY, Chen QZ, Li Y, Shao Y, Ren WY, Liao YP, Wang H, Zhu JH, Huang M, et al: Resveratrol inactivates PI3K/Akt signaling through upregulating BMP7 in human colon cancer cells. Oncol Rep 2017, 38:456-464.

Tomassen MMM, Govers C, Vos AP, de Wit NJW: Dietary fat induced chylomicron-mediated LPS translocation in a bicameral Caco-2cell model. Lipids Health Dis 2023, 22:4.

Hsu CN, Hou CY, Chang-Chien GP, Lin S, Yang HW, Tain YL: Perinatal Resveratrol Therapy Prevents Hypertension Programmed by Maternal Chronic Kidney Disease in Adult Male Offspring: Implications of the Gut Microbiome and Their Metabolites. Biomedicines 2020, 8.

Kranawetter C, Zeng S, Joshi T, Sumner LW: A Medicago truncatula Metabolite Atlas Enables the Visualization of Differential Accumulation of Metabolites in Root Tissues. Metabolites 2021, 11.

      12.The effect of temperatures on resveratrol uptake is not discussed at all in the discussion section.

We have supplemented the discussion section on the effect of temperature on resveratrol uptake.

Temperature changes can significantly affect the fluidity of cell membranes. When the external temperature decreases significantly, the fluidity of the cell membrane also decreases significantly. At low temperatures, enzyme activity in the free cells slows and the fluidity of cell membrane is decreased, and this interferes with the transport mechanisms. When the Caco-2 cells were at 37℃, the transport and uptake of resveratrol were normal. However, when the Caco-2 cells were in a low-temperature (4℃) environment, the transport and uptake of resveratrol significantly decrease. The decrease in temperature not only affects the fluidity of cell membranes, but also affects the vitality of cells. When a cell is at a lower temperature, its vitality is also affected. When cell viability decreases, the transport and uptake of drugs also decrease accordingly. In this study, whether the fluidity of cell membranes and cell viability affect uptake and transport remains a scientific issue that we will continue to explore in the future.

  1. The manuscript has around 35% of similarity. Suggest to revise and reducing the percentage.

We have revised the manuscript and reduced the repetition rate.

Reviewer 2 Report

Some remarks:

- please, avoid unnecessary capitals (e.g. in the Abstract or in L191-197)

-L32: please, reconsider "feedstuff additives"

-L60: please, specify the soruce of resveratrol

- L78: what does "sample extraction" refer to?

-Figures 3, 4, 6, 8 are too small to be read properly. Also, Table 1 can only be read partly.

- Table 1: How do you explain the presence of brominated and fluorinated metabolites?

Minor corrections needed

Author Response

Dear Reviewers,

Thank you for the reviewer’ comments concerning our manuscript entitled "The transport and uptake of resveratrol mediated via glucose transporter 1 and its antioxidant effect in Caco-2 cells" (Manuscript ID molecules-2425652). Those comments were all valuable and very helpful for revising and improving our paper, as well as for providing important guiding significance to our study. We have studied the comments carefully and have made corrections, which we hope will meet with your approval. The revised text is marked in yellow in the revised version of the manuscript. The main corrections made to the paper and the responses to the reviewer’ comments with point by point are as follows:

Reviewer #2:

Comments and Suggestions for Authors

Some remarks:

  1. please, avoid unnecessary capitals (e.g. in the Abstract or in L191-197)

Thank you very much for your suggestion. We have made modifications to the non-standard uppercase letters in the manuscript.

  1. L32: please, reconsider "feedstuff additives"

We have removed the feedstuff additives in line 32.

  1. L60: please, specify the soruce of resveratrol

Resveratrol (purity greater than 99%) was purchased from Merck (Darmstadt, Germany).

  1. L78: what does "sample extraction" refer to?

The extraction of samples refers to the extraction of resveratrol from the buffer solution of the Caco-2 cell model in the line 78.

  1. Figures 3, 4, 6, 8 are too small to be read properly. Also, Table 1 can only be read partly.

We have enlarged Figures 3, 4, 6, and 8, and fully displayed Table 1.

  1. Table 1: How do you explain the presence of brominated and fluorinated metabolites?

After careful inspection, we found that our liquid chromatography-mass spectrometer may have been interfered with by previous experiments, resulting in incorrect detection of tetrabromobisphenol A and trifluoroacetic acid. To ensure the accuracy of metabolomics results, we carefully evaluated and reviewed the results, removing some metabolites that do not belong to the cellular metabolic system. We also reanalyzed differential metabolites, discussed and enriched metabolic pathways.

Round 2

Reviewer 1 Report

Thank you for addressing the corrections suggested. I have reviewed your revisions and I am pleased to inform you that I agree with the changes made.